# Upper and Lower Motor Neuron Degenerations Are Somatotopically Related and Temporally Ordered in the *Sod1* Mouse Model of Amyotrophic Lateral Sclerosis

**DOI:** 10.3390/brainsci11030369

**Published:** 2021-03-13

**Authors:** Christine Marques, Thibaut Burg, Jelena Scekic-Zahirovic, Mathieu Fischer, Caroline Rouaux

**Affiliations:** Inserm UMR_S 1118, Centre de Recherche en Biomédecine de Strasbourg, Faculté de Médecine, Université de Strasbourg, 67 000 Strasbourg, France; CMARQUES2@mgh.harvard.edu (C.M.); thibaut.burg@kuleuven.be (T.B.); Jelena.Scekic-Zahirovic@dzne.de (J.S.-Z.); mathieu.fischer@paediatrics.ox.ac.uk (M.F.)

**Keywords:** amyotrophic lateral sclerosis, motor cortex, upper motor neurons, lower motor neurons, somatotopy, descending propagation, dying-forward

## Abstract

Amyotrophic lateral sclerosis (ALS) is a devastating and fatal neurodegenerative disease arising from the combined degeneration of upper motor neurons (UMN) in the motor cortex, and lower motor neurons (LMN) in the brainstem and spinal cord. This dual impairment raises two major questions: (i) are the degenerations of these two neuronal populations somatotopically related? and if yes (ii), where does neurodegeneration start? If studies carried out on ALS patients clearly demonstrated the somatotopic relationship between UMN and LMN degenerations, their temporal relationship remained an unanswered question. In the present study, we took advantage of the well-described *Sod1^G86R^* model of ALS to interrogate the somatotopic and temporal relationships between UMN and LMN degenerations in ALS. Using retrograde labelling from the cervical or lumbar spinal cord of *Sod1^G86R^* mice and controls to identify UMN, along with electrophysiology and histology to assess LMN degeneration, we applied rigorous sampling, counting, and statistical analyses, and show that UMN and LMN degenerations are somatotopically related and that UMN depletion precedes LMN degeneration. Together, the data indicate that UMN degeneration is a particularly early and thus relevant event in ALS, in accordance with a possible cortical origin of the disease, and emphasize the need to further elucidate the molecular mechanisms behind UMN degeneration, towards new therapeutic avenues.

## 1. Introduction

ALS is an incurable and fatal neurodegenerative disease that mostly starts in adulthood and rapidly progresses to paralysis and death within only 2–5 years of diagnosis [1]. In clinics, ALS is defined as the combined degeneration of the upper motor neurons (UMN) whose cell bodies are located in the motor cortex and that extend axons to the medulla and spinal cord, and lower motor neurons (LMN) whose cell bodies are located in the medulla and spinal cord, and that connect to the skeletal muscles. This dual neuronal contribution is particularly relevant given that ALS is the most severe disease of the adult motor system, in comparison with diseases that target UMN or LMN only [2]. Dual impairment also raises two major questions: (i) are the degenerations of these two neuronal populations of the motor system somatotopically related? and (ii) where does neurodegeneration start? This last question is altogether timely and therapeutically relevant, given the overall failure of the solely LMN-oriented research to provide treatment to prevent or lastingly slow down disease progression [3].

Clinical and imaging analyses have now clearly established that UMN and LMN degenerations are somatotopically related in ALS patients [4,5,6,7,8], suggesting a causal effect between the two neuronal depletions. However, determining where neurodegeneration starts may not be possible in patients due to technical limitations: (i) UMN signs may be harder to reveal than LMN signs, and LMN signs may partly mask UMN signs; (ii) while imaging techniques and magnetic resonance imaging (MRI) in particular are perfectly adapted to the study of the brain, they are harder to implement in the spinal cord; (iii) if LMN can be easily identified upon choline acetyltranferase (ChAT) staining in post-mortem tissues, there is still no marker to identify human UMN. Indeed, Betz cells, can be identified based on their size and morphology, were clearly shown to have degenerated in post-mortem tissues from ALS patients [9,10], but the distribution of these extra-large neurons is not proportional to the cortical representation of the different body regions along the motor homunculus [11], and they represent only a subpopulation of UMN. Thus, Betz cells quantification does not allow to assess the real extent of UMN degeneration in human. Consequently, no technique exists that allows neurologists or pathologists to compare the actual extent of UMN and LMN loss in ALS patients at one given time point.

Rodent models may prove useful to compare UMN and LMN degenerations. While clinical examination of small animals is difficult, especially regarding UMN signs, histology can be performed from presymptomatic to disease end stage. Similarly to humans, rodent LMN can be easily detected in the ventral horn of the spinal cord upon ChAT staining, but no unique marker or unique combination of markers exist to identify rodent UMN. Indeed, UMN correspond to a discrete population of subcerebral projection neurons (SubCerPN), the largest excitatory projection neurons of the cerebral cortex. SubCerPN are found in the layer V of the cerebral cortex throughout all cortical areas. UMN instead are found only in the motor and somatosensory cortices, intermixed with other SubCerPN. As of today, genetic or immunostaining allow at best to discriminate layer V SubCerPN from other cortical neurons [12], but not to distinguish UMN from the rest of the SubCerPN. In the absence of a unique molecular marker or unique combination of markers, the only way to identify true UMN is based on their unique axonal projection to the spinal cord, using spinal injections of retrograde tracers.

Quantification of SubCerPN from the rodent motor and somatosensory areas upon staining or reporter gene expression, or quantification of retrogradelly labelled UMN, have already demonstrated the degeneration of these populations in mouse models of ALS [13,14]. Loss of layer V CTIP2-positive SubCerPN and loss of large layer V SubCerPN of the motor cortex were reported in disease end-stage *hPFN1^G118V^* [15] and *C9-BAC* mice [16] respectively. Loss of layer V Thy1-YFP-positive SubCerPN in the motor and somatosensoy cortex of *TDP43^A315T^* mice [17] was described at symptom onset. Early and progressive degeneration of retrogradelly-labelled UMN was initially demonstrated in the *SOD1^G93A/G1H^* mice [18] followed by the *SOD1^G93A^* mice [19] and the floxed *SOD1^G37R^* [20]. Together, these studies indicate that the mouse strains that best mimic ALS symptoms recapitulate UMN degeneration and that proper identification of pure UMN allows detecting early loss of this discrete neuronal population. However, how far do these models recapitulate the somatotopic link between UMN and LMN degeneration, and what can they teach us about the temporal relationship of their loss remain unanswered questions.

To better characterize the temporal and somatotopic relationships between LMN and UMN degenerations, we used the *Sod1^G86R^* mice that typically recapitulate ALS onset and progression, with first motor symptoms detected at around 90 days of age and premature death at around 110 days [21]. Retrograde labelling from the cervical or lumbar portions of the spinal cord, followed by rigorous sampling and counting of UMN and LMN from the same animals and application of similar statistical tests, demonstrated that UMN depletion starts long before LMN depletion, and that UMN and LMN degenerations are somatotopically related, highlighting the earliness and relevance of UMN degeneration in ALS.

An initial report of the findings presented here was published as a preprint on bioRxiv [22].

## 2. Material and Methods

### 2.1. Animals

All animal experiments were performed under the supervision of authorized investigators and approved by the local ethics committee of Strasbourg University (CREMEAS, agreements #00766.01 and 1534-2015082617403748). Bacterial artificial chromosome (BAC) transgenic male mice with the G86R murine *Sod1* missense mutation [21] were obtained from the animal facility of the Faculty of Medicine, University of Strasbourg. Non-transgenic age-matched male littermates served as controls. Mice received water and regular rodent chow ad libitum. *Sod1^G86R^* animals were followed daily, and disease progression was rated according to a clinical scale going from score 4 to 0, as we previously described [23]. End-stage animals were harvested upon reaching score 0, i.e., when they were no longer able to roll over within 10 s after being gently placed on their back [23].

### 2.2. Retrograde Labelling of UMN

Mice were deeply anesthetized with intraperitoneal injection of Ketamine (Imalgène 1000^®^, Merial; 120 mg/kg body weight) and Xylazine (Rompun 2%^®^, Bayer; 16 mg/kg body weight) solution and placed on a heating pad. A precise laminectomy was performed in cervical (C3–C4) or lumbar (L1–L2) region of the spinal cord, depending on the experiment, and the animals were positioned below an injector (Nanoject II, Drummond Scientific, Broomall, PA, USA) mounted on a micromanipulator. Using pulled glass capillaries, the dura was punctured, the capillary lowered to the dorsal funiculus, and five pressure microinjections of 23 nL of Fluorogold (Fluorchrome, Denver, CO, USA) were performed on each side of the midline. Five days after Fluorogold injection, or upon reaching disease end stage, the animals were transcardially perfused with cold 0.01 M phostphate buffer saline (PBS), followed by cold 4% paraforlmaldehyde (PFA) in 0.01 M PBS and their brains, spinal cords, tibialis anterior, and gastrocnemius muscles were collected.

### 2.3. UMN Quantification

Brains from Fluorogold (FG) injected animals were cut in 40 μm-thick coronal sections on a Leica VT 1000S vibratome (Leika Biosystems, Wetzlar, Germany) and mounted on slides with DPX mounting solution (Sigma-Aldrich, Saint-Louis, MO, USA). Sections were imaged under an AxioImager.M2 fluorescence microscope equipped with a structured illumination system (Zeiss, Oberkochen, Germany) and a highly sensitive black and white camera (Hamamatsu Photonics, Hamamatsu City, Japan). Images were acquired using the ZEN 2 software (Zeiss). For mice injected in the cervical portion of the spinal cord, all FG-positive neurons present on both hemispheres of eight equally spaced coronal sections, separated by 480 μm and spanning from 2.10 to −1.70 mm relative to Bregma, were counted. For mice injected in the lumbar portion of the spinal cord, all FG-positive neurons present on both hemispheres of three equally spaced coronal sections, separated by 480 μm and spanning from −0.46 to −1.70 mm relative to Bregma, were counted. UMN quantification was performed by two experimenters blinded to the age and genotype of the samples using a macro that we developed on ImageJ (1.48 k, NIH, Bethesda, MD, USA). In brief, the macro generates a reversed image and performs a noise reduction with the “Median filter” (radius = 2). A Gaussian blur (sigma = 25) filtered image is then created and subtracted from the non-blurred image. The result is handled by the “rolling ball radius Subtract Background” algorithm (rolling = 18). The edges of the cells are sharpened and enhanced by the “Unsharp Mask filter” (radius = 50 mask = 0.6). The user then can adjust the automatically set threshold that will be used for the determination of black and white pixels of the binarized image version. The image is then binarized with the “Make Binary” function and subjected to several morphological operations to close (“Close”) and fill holes (“Fill Holes”) in cells and eliminate small objects (“Erode” and “Dilate”) and separate touching cells by “Watershed” segmentation. If needed, the macro allows a manual intervention to further separate touching cells, complete the filling of the holes, and define a ROI around the whole cell population to avoid image border artefacts to be counted. Cells are then counted with the “Analyze particles” function. The macro automatically exports the results as a spreadsheet file and an image file.

The cDNA clone for the mouse *Mu-Crystallin* (*Crym*) is a kind gift from the Arlotta lab. Riboprobes were generated as previously described [24]. Nonradioactive in situ hybridization was performed on 40 μm vibratome coronal brain sections. Selected sections were mounted on superfrost slides and processed using reported methods [24]. Bright field 10X tiles images were acquired with a fluorescence microscope equipped with a Digital Sight DS-U3 camera and run by the Nis Elements 4.0 software (Nikon, Tokyo, Japan). *Crym*-positive neurons were manually counted, on both hemispheres, within layer V, from M2 medially to S1 laterally by two experimenters blinded to the age and genotype of the samples.

### 2.4. Lower Motor Neurons Quantification

The L1–L4 lumbar level of the spinal cords were cut on a vibratome into coronal sections of 40 μm. Four nonadjacent sections spaced by 320 μm were labelled by immunohistochemistry as previously described [25], using a goat anti-choline acetyltransferase (ChAT) antibody (Millipore, Burlington, MA, USA) and a biotinylated donkey anti-goat IgG (Jackson ImmunoResearch, West Grove, PA, USA). Two images per section (one per ventral horn) were captured using an AxioImager. M2 microscope (Zeiss, Oberkochen, Germany) equipped with a high-resolution B/W camera (Hamamatsu Photonics, Hamamatsu City, Japan) and run by the ZEN 2 software (Zeiss). Cell body sizes were measured using ImageJ (NIH, Bethesda, MD, USA).

### 2.5. Electromyography

Mice were anesthetized with a solution of ketamine/xylazine (100 mg/kg; 5 mg/kg) and kept under a heating mat to maintain a physiological muscle temperature (±31 °C). Recordings were performed with a standard electromyographic apparatus (Dantec Dynamics, Skovlunde, Denmark) as previously described to detect electrical activity of the tibialis anterior and gastrocnemius muscles of both limbs for at least 2 min as previously described [23]. For each muscle, a score of 1 (innervated, i.e., not presenting any spontaneous muscle activity) or 0 (denervated, i.e., presenting spontaneous muscle activity) was given, and the total scores of the four muscles were summed.

### 2.6. Neuromuscular Junctions Staining and Morphological Analysis

Tibialis anterior muscles were dissected into bundles and processed for immunofluorescence with a rabbit anti-synaptophysin antibody and a rabbit anti-neurofilament antibody (Eurogentec, Seraing, Belgium) followed by Alexa-conjugated donkey anti-rabbit (Jackson) and rhodamine-conjugated α-bungarotoxin (Sigma), as previously described [25]. Neuromuscular junctions (NMJs) analysis was performed by an independent age and genotype-blinded observer, directly under and AxioImager. M2 microscope (Zeiss) was equipped with a high-resolution B/W camera (Hamamatsu) and run by the ZEN 2 software (Zeiss). On average, 500 NMJs per animal were examined. NMJs were considered partially denervated when the presynaptic nerve terminal labelled with synaptophysin was partially observed from the postsynaptic region labelled with α-bungarotoxin, and denervated when the presynaptic nerve terminal was totally absent. Representative NMJs were imaged using the same microscope setting.

### 2.7. Statistical Analyses

Data are expressed as the means ± SEM (standard error of the mean). Statistical analyses were performed using GraphPad Prism 6 (GraphPad, San Diego, CA, USA). Two-way ANOVA was used to test the overall genotype effect across ages or section numbers, followed by multiple comparisons test using Sidak’s method to assess the genotype effect at each individual age or section number. Multiple *t*-test using Benjamini and Hochberg’s method was employed to analyze EMG scores.

## 3. Results

### 3.1. UMN Progressively Degenerate in Sod1^G86R^ Mice

To test whether *Sod1^G86R^* mice recapitulate UMN degeneration, we performed a series of retrograde labelling of the whole UMN population by injecting Fluorogold (FG) in the cervical portion of the spinal cord of wild type (*WT*) and *Sod1^G86R^* animals (Figure 1A), at two presymptomatic ages—60 and 75 days, and two symptomatic ages—90 and 105 days or end-stage (ES); (Figure 1B). Microscopy analysis of coronal sections revealed labelled pyramidal cells in the layer V of the cerebral cortex, spanning rostro-caudally from Bregma 2.10 to −2.30 mm, and medio-laterally from Interaural 0.5–2 mm (Figure 1C). We then quantified the number of labelled UMN present in both hemispheres of eight coronal sections equally spaced along the rostro-caudal axis and matched between the two genotypes. The results show that the population of retrogradely-labelled UMN progressively and significantly decreased over time in *Sod1^G86R^* animals compared to their *WT* littermates (Figure 1D; *p* < 0.0001 in two-way ANOVA; *p* = 0.0025 at 90 d and *p* < 0.0001 at ES in multiple comparisons test).

To confirm these results and exclude the possibility that the loss of UMN could be due to a defective retrograde transport of FG by the axons of the *Sod1^G86R^* UMN, we compared the proportions of FG-positive neurons (Figure 2A) to that of sub-cerebral projection neurons (SubCerPN, Figure 2B) between *Sod1^G86R^* mice and their *WT* littermates at one single rostro-caudal level: Bregma + 0.74 mm. Within the cerebral cortex, several genes show a specific expression or a highly enriched expression in SubCerPN, such as *Ctip2*, *Fezf2*, or *mu-crystallin* (*Crym*), among others [24]. We chose to reveal *Crym*, that we recently better characterized and whose expression is found in FG-positive UMN [20] among other SubCerPN. We revealed *Crym* expression by in situ hybridization (ISH, Figure 2B). Quantification of the *Crym*-positive neurons present in layer V of the motor and sensory-motor area revealed similar progressive and significant reductions as the quantification of FG-positive neurons (Figure 2C; *p* = 0.0003 in two-way ANOVA and *p* = 0.0001 at disease ES in multiple comparison test for FG-positive neurons, and Figure 2D; *p* = 0.0019 in two-way ANOVA and *p* = 0.0089 at disease ES in multiple comparisons test for *Crym*-positive neurons) in *Sod1^G86R^* mice compared to their *WT* littermates. While the data do not rule out the possibility that retrograde transport may be altered in *Sod1^G86R^* mice UMN, they nevertheless confirm a progressive loss of the UMN population in these animals over time.

### 3.2. UMN Degeneration Propagates in a Caudo-Rostral Manner in Sod1^G86R^ Mice

In mice, UMN are topographically organized within the motor and somatosensory cortices along the rostro-caudal and the medio-lateral axes, and project accordingly to different portions of the spinal cord [26]. Sampling the whole population of UMN along the rostro-caudal axis enabled us to test whether their degeneration was homogeneous, or rather selectively affected sub-populations. We thus quantified the number of UMN present on each of the eight selected coronal sections of *WT* and *Sod1^G86R^* brains at each of the four time points of interest (Figure 3A). Using this approach, we observed that the number of UMN was uneven along the rostro-caudal axis, with a minimal number of neurons found in Section 4 (Bregma +1.18 mm) and a maximum found in Sections 6 and 7 (Bregma +0.14 and −0.46 mm) (Figure 3B–E). At the presymptomatic ages 60 d and 75 d, two-way ANOVA analyses revealed a significant difference between the two genotypes (*p* = 0.0064 at 60 d and *p* = 0.0096 at 75 d; Figure 3B,C), but no difference was seen at the section level using multiple comparisons test. However, from 90 d on, we observed in addition to the overall genotype effect (*p* < 0.0001 both at 90 d and disease end stage in two-way ANOVA), significant reductions of the number of UMN in individual sections. This started at 90 d with the most caudal sections (Section 9: 28.4% loss *p* < 0.001; Section 8: 29.6% *p* < 0.001; Section 7: 19.2% *p* < 0.05; Section 6: 19.2% *p* < 0.01; Figure 3D) and propagated rostrally to reach almost all the sections at disease end stage (section 9: 44.6% loss *p* < 0.001; Section 8: 26.6% *p* < 0.01; Section 7: 20.16% *p* < 0.05; Section 5: 32.77% *p* < 0.01; Section 4: 48.2% *p* < 0.05; Section 2: 68.7% *p* < 0.0001; Figure 3E). Within the caudal part, Section 9 remained the most severely affected, with a loss of 44.68%, but overall, the most impacted section was the rostral Section 2, with a late and abrupt loss of 68.67% (Figure 3E). Together, the data show that UMN degeneration is not even along the rostro-caudal axis in *Sod1^G86R^* mice: it starts with the most caudally located ones and progresses rostrally over time.

### 3.3. Lumbar-Projecting UMN Are Affected Earlier and to a Greater Extent Than the Rest of the UMN Population

Because the most caudal parts of the motor and somatosensory cortices contain the lumbar-projecting UMN [26], we reasoned that this subpopulation of UMN might account for the majority of the neuronal loss observed in the most caudal sections we analyzed. To test this hypothesis, we performed a second set of retrograde labelling and injected Fluorogold in the lumbar portion of the spinal cord of *WT* and *Sod1^G86R^* animals (Figure 4A), at the same presymptomatic and symptomatic ages as previously (Figure 4B). Microscopy analysis of coronal sections revealed labelled cortical cells, spanning from Bregma −0.46 mm to Bregma −1.70 mm (Figure 4C). Sampling and counting the lumbar-projecting UMN revealed a marked loss of this subpopulation in the *Sod1^G86R^* mice compared to their *WT* littermates, which was significant from 75d on, and intensified over time (Figure 4D,E; *p* < 0.0001 in two-way ANOVA; *p* = 0.018 at 75 d and *p* < 0.0001 at 90 d and disease end stage). Taken together, the data show that, in the *Sod1^G86R^* animals, the loss of lumbar-projecting UMN starts earlier than that of the overall UMN population.

### 3.4. UMN and LMN Degenerations Are Somatotopically Related and Temporally Ordered

The paralysis that progressively affects *Sod1^G86R^* mice typically starts in the hind limbs [21]. In accordance with this, lower motor neurons (LMN) were shown to degenerate in the lumbar portion of the spinal cord [21,23]. In order to compare the time course of lumbar-projecting UMN degeneration to that of lumbar LMN degeneration, we counted the number of large, disease-vulnerable ChAT-positive alpha motor neurons (400 µm^2^ or above) present in the ventral horn of the lumbar spinal cord of the same *WT* and *Sod1^G86R^* animals that had received FG-injections (Figure 5A,B). We observed an important and significant loss of large ChAT-positive neurons at end-stage of the disease and a milder and non-significant loss at 90 d (*p* = 0.01 in two-way ANOVA and *p* = 0.0008 at ES in multiple comparisons test, for a final loss of 65.27%; Figure 5B). Together, the data indicate that in the *Sod1^G86R^* ALS mouse model, loss of lumbar LMN cell bodies is a very late event in comparison with the overall disease progression and with the loss of lumbar-projecting UMN cell bodies, which is already significant pre-symptomatically.

In the *SOD1^G93A^* mouse model of ALS, LMN were shown to undergo “dying-back” or Wallerian degeneration, characterized by initial alteration of the neuromuscular junction (NMJ) followed by axonal degeneration and final cell body loss [27]. Electromyographic recordings of fasciculations were shown to precede motor unit instability and reinnervation in ALS patients, and are thus considered one of the earliest marker of LMN denervation [28].To determine when lumbar LMN degeneration starts in the *Sod1^G86R^* mice, we performed electromyography in the gastrocnemius and tibialis anterior muscles (Figure 5C,D) of both hind limbs of *Sod1^G86R^* mice and their control littermates, prior to harvesting. Intact muscles were given a score of 1, while muscles with fasciculations were given a score of 0, and the sum of the scores of the four muscles was calculated for each animal. First fasciculations could be detected at 90 d, but became statistically significant only at disease end stage (*p* = 0.0002 in two-way ANOVA; *p* < 0.0001 at ES in multiple comparisons test; Figure 5D). To confirm these results, we investigated the status of the NMJ of the tibialis anterior muscle by immunofluorescence (Figure 5E,F). In both *WT* and *Sod1^G86R^* animals we observed healthy, innervated NMJ, along with a small proportion of partly denervated NMJ that displayed discrete disconnections of the axonal terminal (Figure 5E,F). The proportion of partly denervated NMJ was stable across ages and genotypes (Figure 5F), indicating that their occurrence likely reflects a mild alteration of the tissue during the process of bundles preparation. Importantly, fully denervated NMJ that had lost their axonal terminal were detected only in *Sod1^G86R^* animals (Figure 5E). This phenotype was significant only at disease end stage (*p* < 0.0001 in two-way ANOVA; *p* < 0.0001 at ES in multiple comparisons test; Figure 5F). Thus, the data indicate that in *Sod1^G86R^* mice, lumbar LMN degeneration starts long after the loss of lumbar-projecting UMN has begun.

### 3.5. UMN Degeneration Is a Particularly Early Event in the Disease Progression of Sod1^G86R^ Mice

ALS is not only characterized by the signs of UMN and LMN degeneration, but also by an important weight loss that often starts decades before appearance of the first motor symptoms [29]. Similarly, *Sod1^G86R^* mice were shown to start losing weight pre-symptomatically, as soon as 75 d of age [30]. To better determine the earliness of UMN degeneration in the course of the disease, we analyzed weight loss in our group of animals and applied similar statistical analyses as for the other parameters (Figure 6). As expected, two-way ANOVA revealed a genotype effect (*p* < 0.0001), and multiple comparisons test unraveled significant differences at 90 d and disease end stage (*p* < 0.0001 in both case), but not at earlier ages, suggesting that weight loss cannot be detected pre-symptomatically in small cohorts of *Sod1^G86R^* mice, as we used in this study.

Taken together, the data show that in a mouse model of ALS that displays initial paralysis in the hind limbs, UMN and LMN degenerations are somatotopically related, with lumbar-projecting UMN being affected earlier than the rest of the UMN population in accordance with initial signs occurring in the hindlimbs. In addition, the data indicate that UMN depletion occurs long before LMN depletion and before the first signs of NMJ denervation. Finally, comparing UMN depletion to weight loss indicates that UMN degeneration is a particularly early event in the disease process. The spatiotemporal dynamics of UMN degeneration during the course of the disease in *Sod1^G86R^* mice contribute to highlight the clinical relevance of UMN and of the motor cortex in ALS.

## 4. Discussion

### 4.1. Somatotopic Relationship between UMN and LMN Degenerations

In clinics, ALS diagnosis relies on evidence of combined degeneration of UMN in the motor cortex, and LMN in the brain stem and spinal cord. This dual impairment nourished a still unresolved debate regarding the potential origin of the neurodegeneration: cortical versus spinal (reviewed in [31]). For Charcot, the lateral sclerosis, i.e., the degeneration of the corticospinal tracts that contain the axons of the UMN and that run laterally in human, was amyotrophic, i.e., responsible for the degeneration of the LMN and the muscular denervation [32]. For Gowers instead, UMN and LMN degenerations were simultaneous but independent [33]. It has now been proven, by different means, that UMN and LMN degenerations are somatotopically related, ruling out the possibility that they can be fully independent from each other. Indeed, clinical examinations of early focal motor manifestations in patients revealed that upper and lower motoneuron signs were maximal in the same peripheral body region at disease onset [34]. Combined magnetic resonance imaging (MRI) and ALS functional rating scale (ALS-FRS), or MRI and electromyography (EMG) have since then confirmed early focal structural changes along the motor homunculus and their correlation with the location of the onsets of LMN signs [4,5,6,7,8]. This correlation between UMN sign/focal cortical changes and the first affected body regions indicates a strong somatotopic link between cortical and spinal impairments observed as early as disease onset. Strikingly, post-mortem analyses revealed a similar correlation between UMN and LMN degeneration [9], suggesting that the initial somatotopic relationship between UMN and LMN impairments may persist throughout the whole disease duration even though it may be harder to observe as disease progresses. This demonstration obtained from clinical, imaging, and histological analyses is impressive, given the heterogeneity of the first clinical presentation of ALS patients and, amongst other parameters, the great variability in the relative severity of UMN and LMN signs, along with the site of disease onset [35].

Thus, demonstrations of early relationship between UMN and LMN dysfunction and degeneration in ALS patients rely on the use of imaging technics, detailed clinical examination, electromyography, and recently implemented techniques of neurophysiology. While some of those techniques exist in rodents, their use remains limited. One of the reasons for this is probably that mouse models allow the use of alternative and complementary sets of techniques, either more invasive or even terminal, that can be applied at any stage of the disease. Thus, histology, which allows investigating the state of any neuronal populations at any age of the animals, has been the preferred technique so far to investigate UMN and LMN degeneration in animal models of ALS. While it does not allow to test for neuronal dysfunction, it reveals actual neuronal loss.

Mutant *SOD1* mouse models typically present with initial hind limb impairment, and so do the *Sod1^G86R^* mice [21,23]. We took advantage of this well characterized model of ALS to test whether mice could recapitulate the somatotopic relationship between UMN and LMN impairments. Our data show that *Sod1^G86R^* mice displayed progressive depletion of the overall UMN population and that UMN loss started caudally, with lumbar-projecting UMN, and progressed rostrally. This selectivity is in accordance with the selectivity of the first LMN signs in this mouse line. Together, our data indicate that UMN and LMN degenerations occur in a somatotopic manner in *Sod1^G86R^* mice, with UMN and LMN involved in hind limb motor control being affected prior to the other subpopulations. Thus, taking into account the distinct tool boxes to assess neuronal dysfunction or cortical thinning in patients on the one hand, and neuronal loss in mice on the other hand, our data indicate that *Sod1^G86R^* mice recapitulate the somatotopic relationship between UMN and LMN impairments reported for ALS patients, highlighting the relevance of this animal model to study UMN and cortical contribution to ALS.

### 4.2. Temporal Relationship between UMN and LMN Degenerations

Focal thinning of the motor cortex of ALS patients at disease onset [4,5,6,7,8] is a strong indication of the presymptomatic degeneration of cortical neurons. It is indeed reasonable to assume that before the thinning of the motor cortex can be detected by MRI techniques, numerous neurons have been lost, and that before cortical neuron cell bodies have disappeared, numerous cortical neurites have degenerated. In other words, significant thinning of the motor cortex at disease onset in ALS patients likely testifies to a long-term process initiated months or years before the appearance of the first LMN-related symptoms. In addition, important neurophysiological investigations, notably by the Australian groups of M. Kiernan and S. Vucic, clearly established an early impairment of the motor cortex in ALS (reviewed in [31]). Indeed, advanced transcranial magnetic stimulation (TMS) techniques demonstrated that both sporadic and familial ALS patients present with early cortical hyperexcitability, characterized by increased intracortical facilitation and decreased short intra cortical inhibition [31]. Importantly, cortical hyperexcitability precedes LMN signs and negatively correlates with survival [31]. While altered excitability of the motor cortex in ALS has not yet been correlated with actual neuronal loss, these studies nevertheless underline the earliness of cortical impairment in the disease process.

The somatotopic relationship between UMN and spinal LMN degenerations that we unraveled in the *Sod1^G86R^* mice allowed us to further investigate their temporal relationship and to determine whether neurodegeneration, in these animals, follows a descending path, from the motor cortex to the spinal cord (“dying forward hypothesis”), or an ascending path, from the spinal cord to the motor cortex (“dying back hypothesis”). Our data demonstrate that the depletion of the UMN cell bodies within the motor cortex started long before the depletion of the LMN cell bodies within the spinal cord, and even before the very first signs of LMN degeneration, i.e., NMJ denervation. The data are in accordance with previous reports that described UMN population depletion prior to motor onset in the *SOD1^G93A/G1H^* [18] and the *SOD1^G93A^* mice [19,36]. Together with these reports, our data support the dying forward hypothesis in ALS. Loss of layer V SubCerPN was also demonstrated in *TDP43^A315T^* mice at disease onset [17], and in *hPFN1^G118V^* [15] and *C9-BAC* mice [16] at disease end stage. Assessing the relative time course of UMN and LMN degenerations in these non-*SOD1* mouse models of ALS would further inform on the relative earliness of UMN degeneration in respect to the genotype, and on the universal aspect of the dying forward degeneration in ALS mouse models.

### 4.3. Degeneration versus Disease Propagation in Mouse Models of ALS

With an earlier degeneration of the UMN compared to LMN, the data support not only the dying forward hypothesis, but also the corticofugal hypothesis in mice. The latter arose from post-mortem histopathological analyses of brains from ALS patients [37,38], and alive diffusion tensor imaging (DTI) studies [39], and suggests that the disease may originate in and propagate from the motor cortex to its direct targets along the projections of the CorticoFugal Projection Neurons (CFuPN). To provide a first experimental evidence of the corticofugal hypothesis, we recently generated a mouse model that ubiquitously expresses the *Sod1^G86R^* transgene, a condition sufficient to develop ALS-like symptoms and premature death, and that entirely lacks UMN and other layer V CFuPN [25]. We demonstrated that absence of this CFuPN population delayed disease onset, reduced weight loss and motor impairment, and increased the survival of *Sod1^G86R^* mice, suggesting that CFuPN carry detrimental signals to their downstream targets [25]. In a follow-up study, we further demonstrated that corticofugal propagation of the disease does likely not follow prion-like mechanisms [20]. Indeed, genetic ablation of the *SOD1^G37R^* transgene selectively from UMN and related CFuPN was sufficient to fully prevent UMN degeneration, but not to affect disease onset and survival. In addition, we did not observe any misfolded SOD1^G37R^ aggregates in the cerebral cortex, as opposed to the spinal cord, and we did not measure decreased levels of misfolded SOD1^G37R^ in the spinal cord upon removal of the *SOD1^G37R^* transgene from the UMN, suggesting that the *SOD1^G37R^* mouse model of the ALS misfolded SOD1^G37R^ protein and related aggregates do not propagate in a prion-like manner [20]. Together, these studies suggest that the detrimental signal carried by UMN and related CFuPN to their downstream targets do not rely on prion-like mechanisms. On the other hand, in the *Sod1^G86R^* mouse model of ALS (the present study) and in the *SOD1^G37R^* mouse model [20], we have shown that despite the early onset of UMN loss, about two-thirds of the population remain present at disease end-stage, indicating the existence of both disease-resistant and disease-vulnerable UMN. We thus propose that, in animal models of ALS, UMN contribution is dual: (i) the degeneration of disease-sensitive UMN contributes to the development of hypereflexia and spasticity [25] and (ii) the maintenance of disease-resistant UMN, along with other CFuPN, may continue to carry a descending toxic message from the motor cortex to the spinal cord [13]. Work is in progress to unravel the mechanisms responsible for UMN degeneration and CFuPN toxicity to their targets [22].

## 5. Conclusions

Together with former reports recently reviewed [13,14], the present work contributes to shedding light on the earliness of UMN degeneration in ALS mouse models. The temporal and somatotopic relationship between UMN and LMN degenerations further emphasize the relevance of UMN and of the motor cortex in ALS onset and progression [25], and the need to investigate the molecular mechanisms behind their degeneration and corticofugal propagation of the disease [20,22], towards new avenues for therapeutic intervention in ALS.

## Figures and Tables

**Figure 1 brainsci-11-00369-f001:**
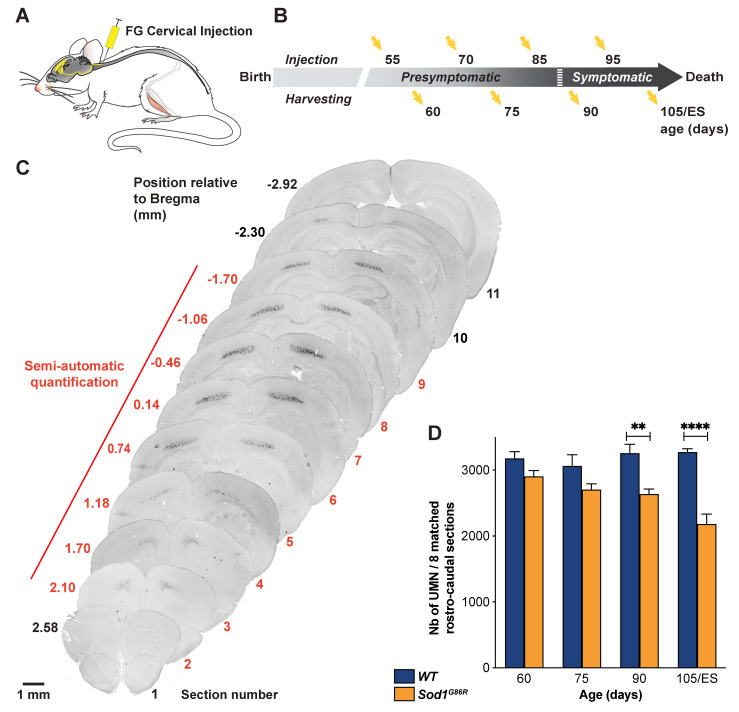
UMN progressively degenerate in *Sod1^G86R^* mice. (**A**) Schematic of the experimental design: UMN were retrogradely labelled upon Fluorogold (FG) injection into the cervical portion of the spinal cord. (**B**) Ages of injection and harvesting (ES: End Stage). (**C**) Representative negative fluorescence images of brain coronal sections numbered 1 (Bregma 2.58 mm) to 10 (Bregma −2.30 mm) showing FG-labelled UMN in the cerebral cortex of a 75 day-old WT mouse. Upon FG injection into the cervical portion of the dorsal funiculus, labelled UMN were consistently observed from section 2 (Bregma 2.10 mm) to 9 (Bregma −1.70 mm). Red numbering and line indicate the sections that were further processed for UMN quantifications. (**D**) Bar graph representing the average number of UMN present on 8 equally spaced coronal sections along the rostro-caudal axis, matched between *Sod1^G86R^* (orange) and *WT* mice (blue). Note the progressive loss of labelled UMN in the brain of *Sod1^G86R^* animals. N= 5 *WT* and 6 *Sod1^G86R^* at 60 days; 6 *WT* and 3 *Sod1^G86R^* at 75 days; 5 *WT* and 6 *Sod1^G86R^* at 90 days; 5 *WT* and 5 *Sod1^G86R^* at 105 days/ES. ** *p* < 0.01 and **** *p* < 0.0001 in multiple comparisons test. Scale bar = 1 mm.

**Figure 2 brainsci-11-00369-f002:**
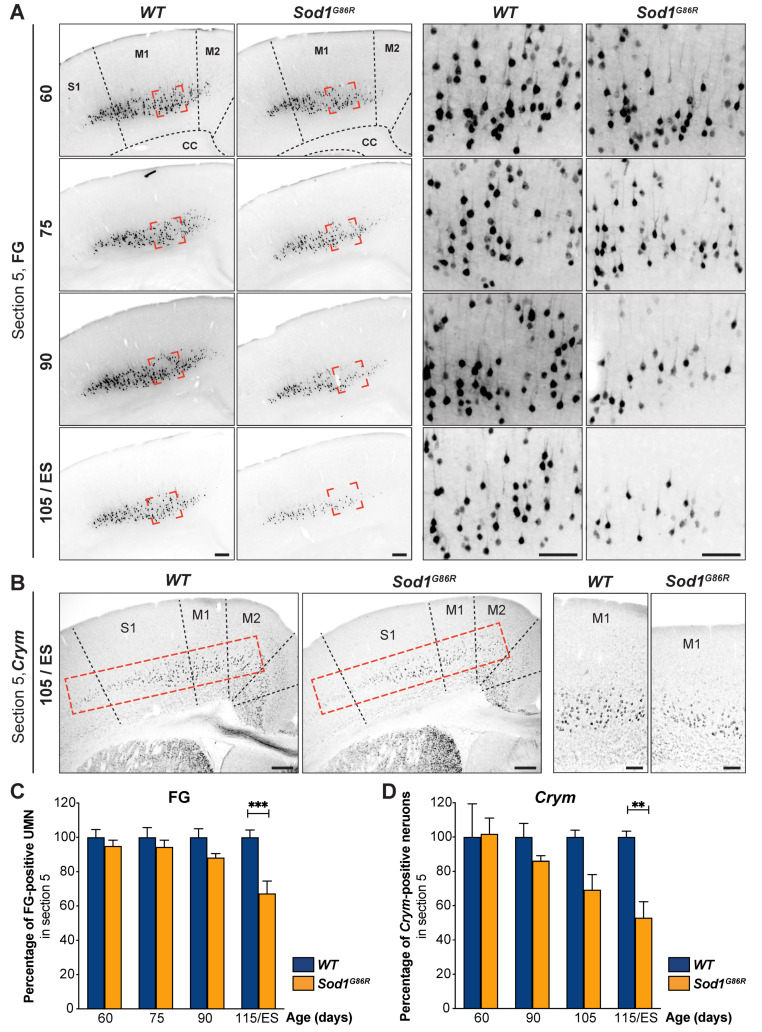
Progressive loss of UMN is confirmed by in situ hybridization. (**A**) Representative negative fluorescence images of brain coronal sections (Bregma 0.74 mm, corresponding to Section 5 for Figure 1) showing FG-labelled UMN in the cerebral cortex of *WT* and *Sod1^G86R^* mice of 60, 75, 90, and 105 days of age or end stage (ES). Red shapes indicate the positions where the close-ups of the right panels were acquired. (**B**) Representative images of in situ hybridization on brain coronal sections of ES *Sod1^G86R^* and 115 day-old *WT* control mice showing decreased *Crym* expression in the layer V of the cerebral cortex. Dotted red rectangles indicate the areas where *Crym*-positive neurons were counted. Close-ups are shown in the right panel. (**C**) Bar graph representing the percentage of FG-positive neurons in *Sod1^G86R^* mice (orange) relative to their *WT* littermates (blue) over time. (**D**) Bar graph representing the percentage of *Crym*-positive neurons in *Sod1^G86R^* mice (orange) relative to their *WT* littermates (blue) over time. N = 5 *WT* and 6 *Sod1^G86R^* at 60 days; 6 *WT* and 3 *Sod1^G86R^* at 75 days; 5 *WT* and 6 *Sod1^G86R^* at 90 days; 5 *WT* and 5 *Sod1^G86R^* at 105 days/ES (end stage). ** *p* < 0.01 and *** *p* < 0.001 in multiple comparisons test. Scale bars = 200 µm in the left panels and 50 µm in the right panels of (**A**,**B**). M1, primary motor cortex; M2, secondary motor cortex; S1, primary somatosensory cortex.

**Figure 3 brainsci-11-00369-f003:**
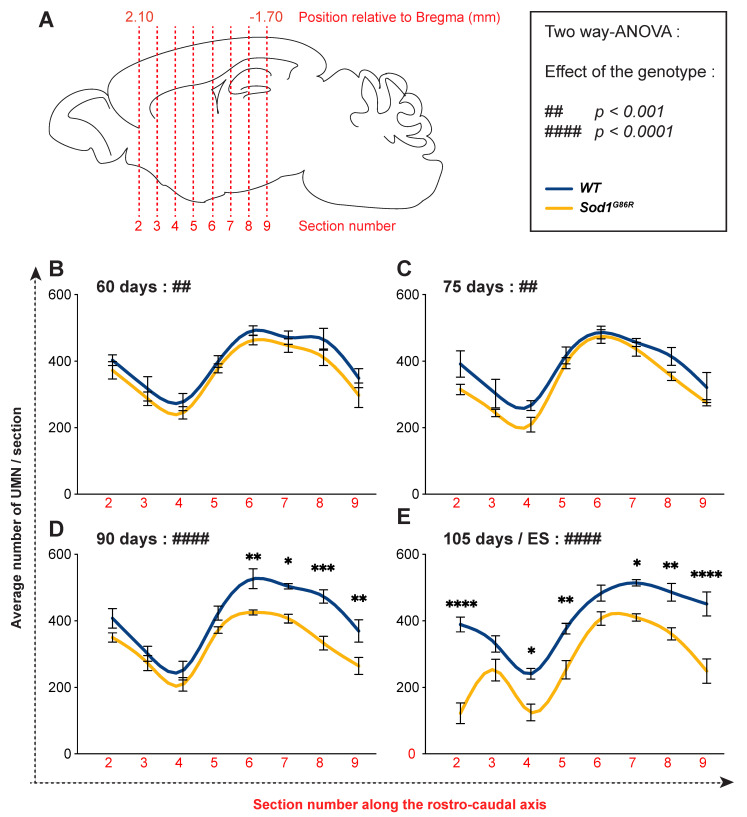
In *Sod1^G86R^* mice, UMN degeneration starts caudally and progresses rostrally. (**A**) Schematic of a mouse brain representing the positioning of the eight rostro-caudal sections where UMN were counted. (**B**–**D**) Graphs representing the average number of FG-labelled UMN counted on both hemispheres of equally spaced coronal Sections 2 (Bregma 2.10 mm) to 9 (Bregma −1.70 mm) of *Sod1^G86R^* mice (orange) and their *WT* littermates (blue) at 60 (**B**), 75 (**C**), 90 (**D**), and 105 days of age or disease end stage (**E**). N = 5 *WT* and 6 *Sod1^G86R^* at 60 days; 6 *WT* and 3 *Sod1^G86R^* at 75 days; 5 *WT* and 6 *Sod1^G86R^* at 90 days; 5 *WT* and 5 *Sod1^G86R^* at 105 days/ES. The box indicates the *p* value obtained for the genotype effect upon running a two-way ANOVA at each age. ## *p* < 0.01 and #### *p* < 0.0001. * *p* < 0.05, ** *p* < 0.01, *** *p* < 0.001 and **** *p* < 0.0001 in multiple comparisons test.

**Figure 4 brainsci-11-00369-f004:**
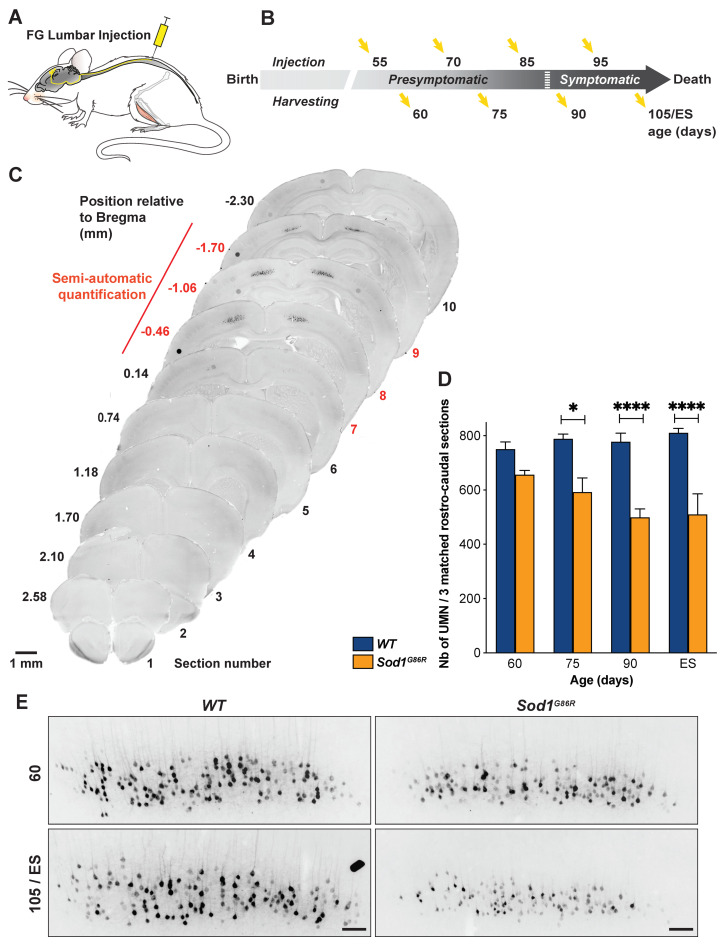
Lumbar-projecting UMN degenerate earlier than the rest of the UMN population. (**A**) Schematic of the experimental design: UMN were retrogradely labelled upon Fluorogold (FG) injection into the lumbar portion of the spinal cord. (**B**) Ages of injection and harvesting. (**C**) Representative negative fluorescence images of brain coronal sections numbered 1 (Bregma 2.58 mm) to 10 (Bregma −2.30 mm) showing FG-labelled UMN in the cerebral cortex of a 75 day-old WT mouse. Upon FG injection into the lumbar portion of the dorsal funiculus, labelled UMN were consistently observed from Section 7 (Bregma −0.46 mm) to 9 (Bregma −1.70 mm). Red numbering and line indicate the sections that were further processed for UMN quantifications. (**D**) Bar graph representing the average number of UMN present on three equally spaced caudal coronal sections matched between *Sod1^G86R^* (orange) and *WT* mice (blue), at 60, 75, 90, and 105 days of age or end stage (ES). N = 4 *WT* and 3 *Sod1^G86R^* at 60 days; 4 *WT* and 3 *Sod1^G86R^* at 75 days; 5 *WT* and 5 *Sod1^G86R^* at 90 days; and 4 *WT* and 4 *Sod1^G86R^* at 105 days/ES (end stage). Note the early and progressive loss of labelled UMN in the brain of *Sod1^G86R^* animals. (**E**) Representative negatives images of fluorescence of brain coronal sections (Bregma −1.06 mm) showing the decreased number of FG-labelled UMN in the cerebral cortex of *Sod1^G86R^* mice at 60 days of age and at end stage (ES) compared to their WT littermates. * *p* < 0.05 and **** *p* < 0.0001 in multiple comparisons test. Scale bar = 1 mm in (**C**) and 100 µm in (**E**).

**Figure 5 brainsci-11-00369-f005:**
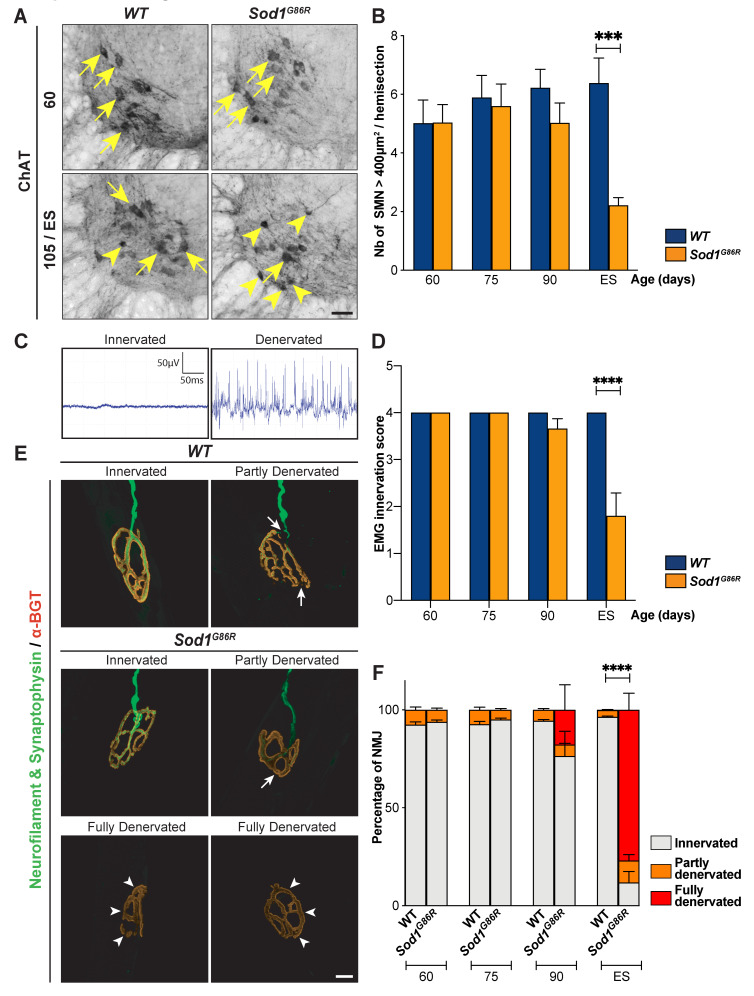
In *Sod1^G86R^* mice, LNM degeneration starts after the loss of the first UMN. (**A**) Representative images of coronal sections of the spinal cord of WT and *Sod1^G86R^* mice at 60 and 105 days or end stage (ES), showing Choline AcetylTransferase-positive neurons (ChAT) present in the ventral horn. Arrows indicate large ChAT-positive neurons, arrowheads indicate smaller or shrinked ChAT-positive neurons. (**B**) Bar graph representing the average number of large ChAT-positive neurons, with an area ≥ 400 µm^2^, counted in the ventral horn of lumbar spinal cord sections from *WT* (blue) and *Sod1^G86R^* mice (orange) at 60, 75, 90, and 105 days of age or ES. (**C**) Representative electromyography recordings of an intact, innervated muscle (left), and of a denervated muscle presenting fasciculations (right). (**D**) Bar graph representing the EMG innervation scores calculated for *Sod1^G86R^* and *WT* mice. (**E**) Representative maximum intensity projection images of z-stacks of typical innervated, partly or fully denervated neuromuscular junctions (NMJ) from 90 day-old *Sod1^G86R^* and *WT* mice upon labelling of presynaptic elements with neurofilament and synaptophysin (green) and postsynaptic elements with alpha-bungarotoxin (α-BGT, red). (**F**) Bar graph representing the average proportions of innervated (grey), partly denervated (orange), or fully denervated (red) NMJ upon staining and microscopy analysis of one tibialis anterior muscle of *WT* and *Sod1^G86R^* mice at 60, 75, 90, and 105 days of age or end stage (ES). N = 5 *WT* and 6 *Sod1^G86R^* at 60 days; 6 *WT* and 3 *Sod1^G86R^* at 75 days; 5 *WT* and 6 *Sod1^G86R^* at 90 days; and 5 *WT* and 5 *Sod1^G86R^* at 105 days/ES. *** *p* < 0.001 and **** *p* < 0.0001 in multiple comparisons test (**B**,**D**) and multiple *t* tests (**F**). Scale bar = 50 µm in (**A**) and 10 µm in (**E**).

**Figure 6 brainsci-11-00369-f006:**
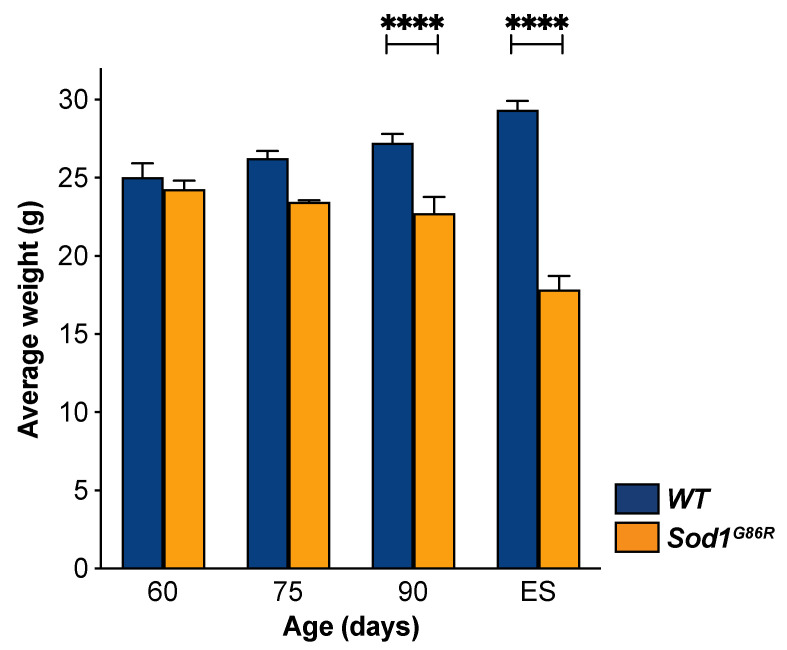
UMN depletion starts before weight loss. Bar graph representing the average weight of animals that received Fluorogold in the cervical or lumbar portions of the spinal cord. Weight was measured prior to harvesting at 60, 75, 90, and 105 days of age or ES. N = 9 *WT* and 8 *Sod1^G86R^* at 60 days; 10 *WT* and 6 *Sod1^G86R^* at 75 days; 10 *WT* and 11 *Sod1^G86R^* at 90 days; and 8 *WT* and 8 *Sod1^G86R^* at 105 days/ES. **** *p* < 0.001 in multiple comparisons test.

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
