# Peer review of "Upper and Lower Motor Neuron Degenerations Are Somatotopically Related and Temporally Ordered in the Sod1 Mouse Model of Amyotrophic Lateral Sclerosis"

_brainsci, 2021, doi:10.3390/brainsci11030369_

Round 1

Reviewer 1 Report

The authors tried to get answers on their questions (first affected structure, temporal and somatopic relationships, etc). Most of their informations were based on animal experiments.  Very few information tackles neurophysiology and clinics. The paper is difficult to read for most clinicicians. My comments:

1.EMG - only denervations (but if only fibs or also fasciculations?), and nothing about MUPs (neurogenic? stabile?..)

2.Correlation to neurophysiology - e.g. short interval cortical inhibition (MEP)

3.Discussion - non-prion-like spread - but their suggestion (or findings) are not clear

4.References - 47 and only 8 of them are really recent.

5.Schemes and graphs - difficult to disclose the message

6.List of abbreviations is missing

Author Response

Specific replies to Reviewer #1:

The authors tried to get answers on their questions (first affected structure, temporal and somatopic relationships, etc). Most of their informations were based on animal experiments.  Very few information tackles neurophysiology and clinics. The paper is difficult to read for most clinicicians.”

We understand that the manuscript, as initially written, may have been difficult to read for clinicians. We think that this may be partly due to the fact that, above the obvious differences that exist between rodents and humans, one of the difficulties may be that basic research and clinics use different sets of tools to investigate neuronal dysfunction and neurodegeneration. This aspect is now the topic of a new paragraph of the discussion.

In addition, to make the reading easier, we have chosen to use solely the terms upper motor neurons (UMN) and lower motor neurons (LMN) and no longer those of corticospinal neurons (CSN) and motorneurons (MN).

 “1.EMG - only denervations (but if only fibs or also fasciculations?), and nothing about MUPs (neurogenic? stabile?..)”

We realize that the text was not describing properly what we recorded with EMG, and why we think this is an early and relevant marker of LMN degeneration. We recorded fasciculations, that were shown to be amongst the first signs of LMN impairment that can be recorded by EMG, revealing initial denervation prior to compensatory effects (which can be assessed by motor unit investigations) (de Carvalho et al., 2017). The corresponding paragraphs of the Methods (p9) and Results (p13) sections have been modified, and we hope this now better justifies the experimental choices.

“2.Correlation to neurophysiology - e.g. short interval cortical inhibition (MEP)”

The reviewer is right underlining that neurophysiology arguments were missing from the initial version of our manuscript. We have now added a paragraph in the discussion part summarizing the large amount of work that had been done to investigate motor cortex dysfunctions in ALS (p17-18).

“3.Discussion - non-prion-like spread - but their suggestion (or findings) are not clear”

We have further developed this part, summarizing the article that was cited (p19), and we hope it is now more clear.

“4.References - 47 and only 8 of them are really recent.”

In the first version of the manuscript 8/47 references were from 2019 or later (17.0%). While trying to keep a bit of the historical perspective regarding the investigations of actual UMN loss in ALS, we have now replaced some references by more recent ones, and removed old publications, bringing the number of recent publications to 10/39 (26.6%).

“5.Schemes and graphs - difficult to disclose the message”

While we were not sure about what really were the difficulties with the figures, we tried to improve them. More specifically, we detailed the graphical abstract and made modifications in Fig.1B, C, D; Fig.2A, B, C, D: Fig3.A; Fig. 4B, C, D; Fig. 5B, D, F and Fig.6.

In particular, we have removed the #, ##, ## and #### symbols from all figures, except for Fig.3 were they are now better explained. These represented the p value of the genotype effect assessed across ages (or section numbers, depending on the experiment) in two-way ANOVA, while * symbols represented the genotype effect assessed age by age (or section by section, depending on the experiment). The p values of the two-way ANOVA analyses can still be found in the text of the Results section.

“6.List of abbreviations is missing”

Abbreviations have been added and can be found p3 of the manuscript.

Reviewer 2 Report

In this paper, the authors used the Sod1G86R mice, which recapitulate ALS onset and progression, to demonstrate that CSN depletion starts long before MN depletion, and that CSN and MN degenerations are somatotopically related. Their results underline the earliness and relevance of CSN degeneration in ALS.

 Data presented by the authors are convincing and experimentally compelling, and, on the whole, the paper significantly increases our knowledge of neurodegenerative process in ALS.

However, the manuscript shows some issues that need to be cleared.

  • My main concern (and the reason for which I have suggested “major” revisions) is about the confusion between mouse model and ALS patients, in particular in the Discussion (e.g. lines 478-482). The authors state that the Sod1G86R mice recapitulate some aspects of human disease, but this does not mean that the conclusions may be directly transferred to ALS patients. The two aspects should be kept distinct. The Sod1G86R mice, however good, are still an animal model.
  • In addition, the statement that this mouse model is similar to SOD1 patients because these mice typically present with initial hind limb impairment is at the very least simplistic. The authors may speculate about the implications of their results on ALS in humans, but the only experimentally proven conclusions are those of the last paragraph of the results (lines 385-393), the other assertions are basically conjectures, and should be presented as such.

Thus, I recommend a significant revision of the Discussion.

Minor points:

  • Please, briefly explain why the mu-Crystallin gene (Crym) has been chosen for in situ hybridization;
  • Insert the meaning of ISH abbreviation in the text;
  • The meaning of the symbols "##" and "###" in the Figures is not very clear. Are there referred to the p value of the whole experiment?

Author Response

Specific replies to Reviewer #2:

Major points

“- My main concern (and the reason for which I have suggested “major” revisions) is about the confusion between mouse model and ALS patients, in particular in the Discussion (e.g. lines 478-482). The authors state that the Sod1G86R mice recapitulate some aspects of human disease, but this does not mean that the conclusions may be directly transferred to ALS patients. The two aspects should be kept distinct. The Sod1G86R mice, however good, are still an animal model.

- In addition, the statement that this mouse model is similar to SOD1 patients because these mice typically present with initial hind limb impairment is at the very least simplistic. The authors may speculate about the implications of their results on ALS in humans, but the only experimentally proven conclusions are those of the last paragraph of the results (lines 385-393), the other assertions are basically conjectures, and should be presented as such.”

We agree with Reviewer #2 that the fact “that Sod1G86R mice recapitulate some aspects of human disease […] does not mean that the conclusions may be directly transferred to ALS patients”. In the same vein, the comparison of Sod1G86R mice with SOD1-ALS patients was clumsy and has been removed.

We have then substantially rewritten the discussion, separating human and rodent studies, and adding a paragraph that further highlights why patients and animal models cannot be directly compared.

We have also limited the paragraph entitled “Degeneration versus disease propagation” to what we’ve learned from the mice, without trying to speculate to what may be happening in ALS patients. We hope that the text removals and additions, along with the new formulations make the discussion less conjectural and more temperate and appropriate.

Minor points :

“Please, briefly explain why the mu-Crystallin gene (Crym) has been chosen for in situ hybridization; Insert the meaning of ISH abbreviation in the text”

These are now detailed in the Results section, p11.

“The meaning of the symbols "##" and "###" in the Figures is not very clear. Are there referred to the p value of the whole experiment?”

We understand that these symbols #, together with the other ones * are confusing. We have now better define, in the Methods sections p10, the statistical analysis that we carried. # symbols were indeed representing the p value of the genotype effect assessed across ages (or section numbers, depending on the experiment) in two-way ANOVA, while * symbols represented the genotype effect assessed age by age (or section by section, depending on the experiment). To make the figures easier to understand, we removed the # symbols except for Figure 3 where they are further explained. We left the p values of the two-way ANOVA analyses in the text of the Results section.

Round 2

Reviewer 2 Report

The authors have exhaustively replied to my comments and modified the manuscript accordingly.